# A Magnetic Integration Mismatch Suppression Strategy for Parallel SiC Power Devices Applications

Shikai Sun [1,2], Jialin Liu [2], Lei Chen [2,*], Zhenlin Lu [2,*], Yuan Wang [1], Wenhao Yang [3], Yuyin Sun [3] and Hui Guo [3]

1  School of Integrated Circuits, Peking University, Beijing 100871, China; sunshikai@mxtronics.com (S.S.); wangyuan@pku.edu.cn (Y.W.)

2  Beijing Microelectronics Technology Institute, Beijing 100076, China; liujialin@mxtronics.com

3  School of Microelectronics, Xidian University, Xi'an 710071, China; yang_wenhao@stu.xidian.edu.cn (W.Y.); yuyinsun@stu.xidian.edu.cn (Y.S.); guohui@mail.xidian.edu.cn (H.G.)

*  Correspondence: chenleinpu@vip.126.com (L.C.); luzhenlin@mxtronics.com (Z.L.)

**Abstract:** A new magnetic integrated parallel current sharing control method for parallel silicon carbide (SiC) power devices is presented in this article. The problem of the application of parallel connected SiC power devices is analyzed. The coupled inductance method is adopted to solve the problem. Based on the active-back converter, we establish the theoretical model of the coupled inductance, and figure out its working mechanism. The integrated magnetic device is designed based on the working mechanism, and the effectiveness is determined through simulation. A 12 V/10 A output magnetic integrated active-flyback converter prototype is fabricated and tested to verify the strategy. Measurement results show that, with the proposed magnetic integrated method, the mismatch voltage is suppressed to 0.1 V under all load conditions, and the efficiency increases by at most 6.52% under full load conditions.

**Keywords:** SiC power devices; magnetic integrated; parallel current sharing; working mechanism





## 1. Introduction

Silicon Carbide (SiC) material can push the power density and efficiency of semiconductor devices and power systems to higher limits due to its wide band gap, high critical field, and high thermal conductivity [1–3]. With the development of SiC technology, the application of SiC power devices is becoming more and more popular [4,5]. The low switching-loss characteristic facilitates a reduction in power loss and an improvement in working frequency, which leads to the use of smaller passive components and improving power density [6,7].

The development of power electronics demands higher and higher current ratings, which promotes the parallel connection of power devices. When the parallel connection is used, the current imbalance among the paralleled power devices becomes a major concern [8]. The current imbalance is caused by the mismatch in device parameters among the paralleled semiconductors or the mismatch in the parasitic parameters of their corresponding circuits when the circuit layouts are asymmetrical. The condition may result in conduction and switching losses, which may further cause thermal distribution problems.

For the SiC power device applications, the value of ON-resistance is smaller than that of the counterpart Si devices. A little mismatch may lead to a large percentage change. Thus, the SiC power devices are more sensitive to the variation of device parameters in paralleled applications. The current mismatch phenomenon has already appeared in the paralleled applications of SiC devices [9,10]. Ref. [11] analyzed the influence of the variability of device parameters on the current sharing of parallel-connected SiC MOSFETs. Experimental investigations of static and transient current sharing were carried out in ref. [12]. The parallel-connected application of packaged SiC power devices was evaluated

in Ref. [13]. The above articles provide a detailed analysis of the mismatch mechanism of the devices. Many studies have been performed on the imbalance of current suppression. From the view of the study object, the imbalance current suppression method can be classified into three categories: device classification; device operating condition monitoring; and circuit topology [14–19]. The chip screening method is proposed to solve the mismatch introduced by the asymmetric layout [20].

The typical representative of the device classification view is the transfer curve distance coefficient classification criterion proposed in ref. [14]. This paper evaluates the factors of the device characteristics and finds that the transfer characteristic contains the main influences. The strategy realizes the mismatch suppression by weighting the distance coefficients of the device transfer curves. The strategy needs to test every device, which limits its massive applications and universality. The typical representation of the device operating condition monitoring is the SiC MOSFET gate driving scheme with a dynamic current equalization mechanism for over-current protection proposed in Ref. [21]. The scheme realizes the simultaneous turn-on of SiC MOSFETs with different threshold voltages by monitoring the device current cycle by cycle to achieve the mismatch current suppression of the parallel device. However, the strategy needs to add extra devices to suppress the mismatch. The circuit structure route to suppress the mismatch current is typified by a parallel current feedback equal-current resonant converter [22]. The strategy adopts a two-stage structure including an interleaved parallel boost converter and a double magnetically coupled half-bridge LLC resonant conversion. This scheme adds two inductors to realize parallel current equalization, which increases the number of magnetic devices in the converter, as well as the iron and copper losses.

In this paper, a novel magnetic integration strategy is proposed to achieve parallel equalization control without increasing the number and size of the converter cores. To verify the proposed strategy, a prototype converter is designed, fabricated, and tested. The measured results show that power efficiency is enhanced by at most 6.52% in the whole load range.

## 2. Operating Principle of the Proposed Strategy

### 2.1. The Topology Evolution

To solve the problem of the mismatch current distribution of SiC power devices in parallel applications, this paper proposes a control method to solve the problem at the topology level. Its topology evolution is shown in Figure 1. Filter inductance is generally used in parallel to reduce copper losses in high-current applications, as shown in Figure 1a. The parallel SiC power devices and filter inductance decoupled to form different branches, as shown in Figure 1b. Since the impedance of the secondary filter inductance is much larger than the on-resistance of the SiC power device, the influence of the device characteristics on the current distribution is converted into the influence of the filter inductance. The topology introduces magnetic coupling by sharing the common magnetic core to suppress the mismatch current, which is shown in Figure 1c. This topology evolution transforms the SiC power devices mismatch into the inconsistency of the filter inductance and further reduces the influence using coupled inductance. And, the final topology does not increase the number of devices.

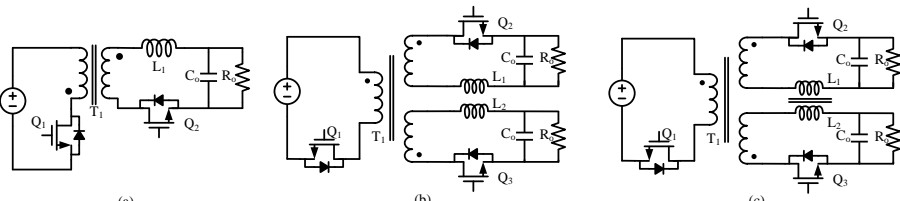

(a)  (b)  (c)

**Figure 1.** Topological evolution process of the parallel equalization control method: (**a**) conventional circuit output flyback converter; (**b**) parallel two-output flyback converter; and (**c**) coupled inductance flyback converter.

### 2.2. The Operational Principle of the Coupled Inductance

There are two main ways of performing coupled inductance, namely flux mutual and flux cancellation. As shown in Figure 2, coil1 corresponds to inductance $L_1$ and has $N_1$ turns whilst coil2 corresponds to inductance $L_2$ and has $N_2$ turns.

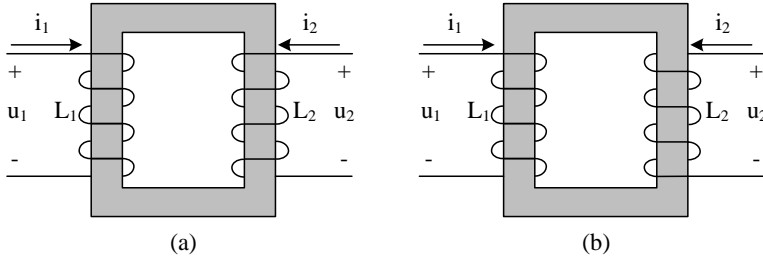

(a)         (b)

**Figure 2.** The two main means of coupled inductance: (**a**) flux mutual; and (**b**) flux cancellation.

$$\Psi_{11} = N_1\Phi_{11} = L_1 i_1 \tag{1}$$

$$\Psi_{21} = N_2\Phi_{21} = M i_1 \tag{2}$$

When a current $i_1$ flows through coil1 in Figure 2a, the total self-induced magnetic flux linkage can be expressed as (1), and the mutual magnetic flux linkage can be expressed as (2), where $\Psi_{11}$ is the self-induced magnetic flux linkage, $\Phi_{11}$ is the mutual magnetic flux generated by cycle of coil1, $L_1$ is the self-induction of coil1, and $\Psi_{21}$ is the mutual magnetic flux linkage generated by coil1 and affecting coil2, $\Phi_{21}$ is the mutual magnetic flux by cycle of coil2, while $M$ is the mutual inductance of coil1 and coil2.

$$\Psi_{22} = N_2\Phi_{22} = L_2 i_2 \tag{3}$$

$$\Psi_{12} = N_1\Phi_{12} = M i_2 \tag{4}$$

Similarly, coil2 also generates a self-induced magnetic flux linkage and mutual magnetic flux linkage. Its self-induced magnetic flux linkage is denoted by $\Psi_{22}$ and its mutual-induced magnetic flux linkage is denoted by $\Psi_{12}$, as $\Psi_{22}$ is the self-induced magnetic flux linkage, $L_2$ is the self-induction of coil2, $\Psi_{12}$ is the mutual magnetic flux linkage generated by coil2 and affecting coil1, $\Phi_{12}$ is the mutual magnetic flux generated by coil2 and affecting coil1, and $M$ is the mutual inductance of coil1 and coil2.

Under linear conditions, $M_{12} = M_{21} = M$, and hereafter $M$ is used to denote mutual inductance. According to the right-handed helix rule, the self- and mutual-inductive flux of the two coils shown in Figure 2a go in the same direction, which is defined as flux mutual, and the total magnetic flux linkage of coil1 and coil2 is denoted by (5) and the port voltage is denoted by (6).

$$\begin{cases} \Psi_1 = \Psi_{11} + \Psi_{12} = L_1 i_1 + M i_2 \\ \Psi_2 = \Psi_{22} + \Psi_{21} = L_2 i_2 + M i_1 \end{cases} \tag{5}$$

$$\begin{cases} u_1 = \dfrac{d\Psi_1}{dt} = L_1\dfrac{di_1}{dt} + M\dfrac{di_2}{dt} \\ u_2 = \dfrac{d\Psi_2}{dt} = L_2\dfrac{di_2}{dt} + M\dfrac{di_1}{dt} \end{cases} \tag{6}$$

The two coils shown in the corresponding Figure 2b have their self-inductive and mutual-inductive fluxes in opposite directions, which is defined as flux cancellation, and the total magnetic flux linkage of coil1 and coil2 is denoted by (7), and the port voltage can be denoted by (8).

$$\begin{cases} \Psi_1 = \Psi_{11} - \Psi_{12} = L_1 i_1 - M i_2 \\ \Psi_2 = \Psi_{22} - \Psi_{21} = L_2 i_2 - M i_1 \end{cases} \tag{7}$$

$$\begin{cases} u_1 = \dfrac{d\Psi_1}{dt} = L_1\dfrac{di_1}{dt} - M\dfrac{di_2}{dt} \\ u_2 = \dfrac{d\Psi_2}{dt} = L_2\dfrac{di_2}{dt} - M\dfrac{di_1}{dt} \end{cases} \tag{8}$$

To simplify the description of the port voltage, the coupling coefficient k is introduced. The coupling coefficient represents the geometric mean of the ratio of mutual inductance to the self-induced inductance chain of the two coils and is expressed by Equation (9).

$$k = \sqrt{\frac{\Phi_{12}\Phi_{21}}{\Phi_{11}\Phi_{22}}} \tag{9}$$

Substituting the magnetic flux linkages separately gives the coupling coefficient expression (10).

$$k = \sqrt{\frac{\Phi_{12}\Phi_{21}}{\Phi_{11}\Phi_{22}}} = \frac{M}{\sqrt{L_1 L_2}} \tag{10}$$

Quantitatively describing coupled coils in terms of coupling coefficients and leakage inductance allows the modelling of coupled coils to be directly embedded in the port voltages of coil1 and coil2, which can be expressed as (11) and (12), respectively.

$$\begin{cases} u_1 = L_{k1}\dfrac{di_1}{dt} + k\sqrt{\dfrac{L_1}{L_2}}u_2 \\ u_2 = L_{k2}\dfrac{di_2}{dt} + k\sqrt{\dfrac{L_2}{L_1}}u_1 \end{cases} \tag{11}$$

$$\begin{cases} u_1 = L_{k1}\dfrac{di_1}{dt} - k\sqrt{\dfrac{L_1}{L_2}}u_2 \\ u_2 = L_{k2}\dfrac{di_2}{dt} - k\sqrt{\dfrac{L_2}{L_1}}u_1 \end{cases} \tag{12}$$

Based on the above analysis, it can be seen that the coupling coefficient is less than or equal to 1, i.e., $k \leq 1$, and the leakage inductance of the two coils can be expressed as $L_{k1} = (1 - k^2)L_1$ and $L_{k2} = (1 - k^2)L_2$, respectively. The coupled inductance voltage-current relationship is reconstructed to create a symmetrical coupled inductance model. Assuming that the coupled inductance has equal values in terms of excitation inductance, Equations (11) and (12) can be expressed as (13) and (14).

$$\begin{cases} L_{k1}\dfrac{di_1}{dt} = u_1 - ku_2 \\ L_{k2}\dfrac{di_2}{dt} = u_2 - ku_1 \end{cases} \tag{13}$$

$$\begin{cases} L_{k1}\dfrac{di_1}{dt} = u_1 + ku_2 \\ L_{k2}\dfrac{di_2}{dt} = u_2 + ku_1 \end{cases} \tag{14}$$

The equivalent circuit of the two coils is shown in Figure 3, where the controlled voltage source represents the coupling effect between the two coils, and the inductance is the respective leakage inductance of the coupled coils. The voltage of the coupled coils in the converter secondary side can be expressed by Equation (15).

$$\begin{cases} u_1 = u_{sec1} - V_{out} \\ u_2 = u_{sec2} - V_{out} \end{cases} \tag{15}$$

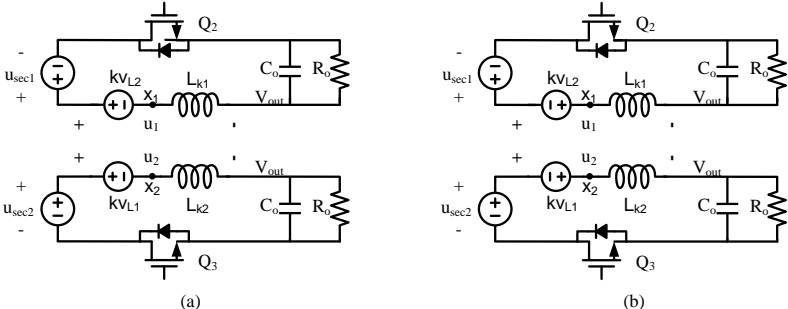

(a)

(b)

**Figure 3.** The equivalent circuit of the two coils: (**a**) flux mutual; and (**b**) flux cancellation.

Since the mismatch is only affected by the secondary side, the following analysis focuses on it. Embedding coupled inductance models into the topological secondary side, the equivalent circuit is shown in Figure 4. The reference points $x_1$ and $x_2$ for the voltages of flux mutual and flux cancellation can be expressed by (16) and (17), respectively.

(a)

(b)

**Figure 4.** The equivalent circuit of the secondary side of the topology: (**a**) flux mutual; and (**b**) flux cancellation.

$$\begin{cases} u_{x1} = u_{sec1} - kv_{L2} = (u_{sec1} - kv_{sec2}) + kV_{out} \\ u_{x2} = u_{sec2} - kv_{L1} = (u_{sec2} - kv_{sec1}) + kV_{out} \end{cases} \tag{16}$$

$$\begin{cases} u_{x1} = u_{sec1} + kv_{L2} = (u_{sec1} - kv_{sec2}) + V_{out} \\ u_{x2} = u_{sec2} + kv_{L1} = (u_{sec2} - kv_{sec1}) + V_{out} \end{cases} \tag{17}$$

Unifying the equivalent voltage source generated by the coupled inductance into the voltage source of the secondary excitation inductance, the complex model of coupled inductance is simplified into the equivalent model of the voltage source and the leakage inductance.

Based on the coupled inductance equivalent model established above, the output current change rate of the converter flux mutual aid and flux cancellation coupled inductance is expressed by Equations (18) and (19).

$$S_F = -\frac{(1-k)V_{out}}{L_k} = -\frac{V_{out}}{(1+k)L} \tag{18}$$

$$S_F = -\frac{(1+k)V_{out}}{L_k} = -\frac{V_{out}}{(1-k)L} \tag{19}$$

The peak value of the current during the steady-state operation of the converter can be obtained according to the converter operating principle, and the peak value of the current for flux mutual and flux cancellation can be expressed by Equations (20) and (21), respectively.

$$\Delta I_{pp} = S_F(1-D)T = \frac{V_{out}}{(1+k)L}(1-D)T \tag{20}$$

$$\Delta I_{pp} = S_F(1-D)T = \frac{V_{out}}{(1-k)L}(1-D)T \tag{21}$$

From the above analysis, it can be seen that the output current ripple suppression effect is positively correlated with the coupling coefficient in the flux mutual; and the output current ripple suppression effect is negatively correlated with the coupling coefficient in the flux cancellation.

The main reasons for the mismatch in current distribution include the mismatch of on-resistance and parasitic inductance at the device level, the passive components at the circuit level, and the parasitic mismatch of the layout. The above mismatches can be expressed by correcting the device model, where $R_{ds}$ denotes the different on-resistance of the two branches and $L_{ds}$ denotes the different parasitic inductance of the two branches. Embedding the modified device model into the output model, the secondary side equivalent circuit of the conventional flyback topology, the flux mutual coupled inductance, and the flux cancellation are shown in Figure 5.

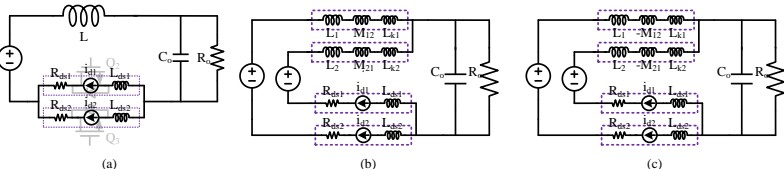

(a)       (b)       (c)

**Figure 5.** The equivalent circuit of the secondary side of the topology: (**a**) without coupled inductance; (**b**) flux mutual; and (**c**) flux cancellation.

Referring to Figure 5a, the mismatch resistance can be expressed as $\Delta R_{ds} = R_{ds1} - R_{ds2}$ and the mismatch inductance can be expressed as $\Delta L_{ds} = L_{ds1} - L_{ds2}$. Since the on-resistance mismatch of the MOSFET is at the mΩ level, its parasitic inductance and that of the circuit layout are at the level of a few nH, while the filtering inductance is at the level of a few tens of μH, and the non-ideal effect can be ignored when performing loop current calculations. Under these conditions, the converter's secondary side current is consistent with the typical current of the converter. The total current at the secondary side in this case can be used in (22).

$$
\begin{aligned}
I_{sec} &= \frac{V_{out}}{sL + (R_{ds1} + sL_{ds1})//(R_{ds2} + SL_{ds2})} \\
&= \frac{V_{out}[R_{ds1} + R_{ds2} + s(L_{ds1} + L_{ds2})]}{sL[R_{ds1} + R_{ds2} + s(L_{ds1} + L_{ds2})] + (R_{ds1} + SLds1)(R_{ds2} + SLds2)}
\end{aligned}
\tag{22}
$$

This current is split between the two branches, and according to Kirchhoff's current law (KCL) and Kirchhoff's voltage law (KVL), the current distribution between the two branches is inversely proportional to the total impedance of the branches, which can be expressed as (23).

$$
\begin{cases}
i_{d1} = I_{sec}\dfrac{R_{ds2} + sL_{ds2}}{R_{ds1} + R_{ds2} + s(L_{ds1} + L_{ds2})} \\[3mm]
i_{d2} = I_{sec}\dfrac{R_{ds1} + sL_{ds1}}{R_{ds1} + R_{ds2} + s(L_{ds1} + L_{ds2})}
\end{cases}
\tag{23}
$$

The mismatch current in the absence of coupling inductance can be expressed by (23), where the parasitic inductance size is roughly at the level of a few nH, while the filtering inductance is in the order of tens to tens of μH and $L \gg L_{ds}$. Therefore, a further simplified representation of the mismatch current can be made.

$$\begin{aligned}
\Delta i &= |i_{d1} - i_{d2}| \\
&= |i_{sec} \frac{\Delta R_{ds} + s\Delta L_{ds}}{R_{ds1} + R_{ds2} + s(L_{ds1} + L_{ds2})}| \\
&= |\frac{V_{out}(\Delta R_{ds} + s\Delta L_{ds})}{sL[R_{ds1} + R_{ds2} + s(L_{ds1} + L_{ds2})] + (R_{ds1} + sL_{ds1})(R_{ds2} + sL_{ds2})}| \\
&\approx |\frac{V_{out}(\Delta R_{ds} + s\Delta L_{ds})}{sL[R_{ds1} + R_{ds2}]}|
\end{aligned} \tag{24}$$

From the above analysis, it can be seen that the current distribution between the two MOSFETs is independent of filter inductance, and only of the resistance and parasitic inductance of the MOSFETs and the layout. The current mismatch is directly determined by the MOSFETs resistance and the parasitic inductance mismatch. Referring to the flux mutual coupling inductance model shown in Figure 5a, the voltage equations for the two branches can be listed as (25) and (26), respectively.

$$V_{Z1} + V_{sec} + V_{L1} + V_{M12} = V_{out} \tag{25}$$

$$V_{Z2} + V_{sec} + V_{L2} + V_{M21} = V_{out} \tag{26}$$

where $V_{Z1}$ and $V_{Z2}$ represent the voltage drop across the MOSFET, which can be expressed by Equation (27).

$$\begin{cases} V_{Z1} = i_{d1} \cdot (R_{ds1} + sL_{ds1}) \\ V_{Z2} = i_{d2} \cdot (R_{ds2} + sL_{ds2}) \end{cases} \tag{27}$$

$V_{sec}$ denotes the equivalent voltage source generated by the transformer coupling to the secondary side, which is determined by the converter parameters and is a constant in steady-state operation. $V_{L1}$ and $V_{L2}$ denote the voltage drops generated by the coupled excitation inductance and leakage inductance, respectively, which can be expressed by the Equation (28).

$$\begin{cases} V_{L1} = L_1 \frac{di_{d1}}{dt} \\ V_{L2} = L_2 \frac{di_{d2}}{dt} \end{cases} \tag{28}$$

$V_{M12}$ denotes the voltage drop corresponding to the mutual inductance generated by inductance $L_2$ over inductance $L_1$, and $V_{M21}$ denotes the voltage drop corresponding to the mutual inductance generated by inductance $L_1$ over inductance $L_2$, which can be expressed by (29).

$$\begin{cases} V_{M12} = M_{12} \frac{di_{d2}}{dt} \\ V_{M21} = M_{21} \frac{di_{d1}}{dt} \end{cases} \tag{29}$$

Since the coupled inductance is wound by the PCB, its consistency and symmetry are extremely high, and the difference generated by the leakage inductance is negligible compared with the excitation inductance and mutual inductance, so it can be assumed that $L_1 = L_2 = L$ and $M_{12} = M_{21} = M$.

Substituting (28) and (29) into Equations (25) and (26) yields (30) and (31).

$$V_{Z1} + V_{sec} + L\frac{di_{d1}}{dt} + M\frac{di_{d2}}{dt} = V_{out} \tag{30}$$

$$V_{Z2} + V_{sec} + L\frac{di_{d2}}{dt} + M\frac{di_{d1}}{dt} = V_{out} \tag{31}$$

Subtract (30) from (31) using the formula (32).

$$V_{Z1} - V_{Z2} + (L - M)(\frac{di_{d1}}{dt} - \frac{di_{d2}}{dt}) = 0 \tag{32}$$

Then, Equation (32) can be reduced to (33).

$$\frac{di_{d1}}{dt} - \frac{di_{d2}}{dt} = \frac{V_{Z1} - V_{Z2}}{M - L} \tag{33}$$

By defining $\Delta i = i_{D1} - i_{D2}$ according to the difference subtraction relation, we can denote (33) by (34).

$$\frac{d\Delta i}{dt} = \frac{d(i_{d1} - i_{d2})}{dt} = \frac{V_{Z1} - V_{Z2}}{M - L} \tag{34}$$

Due to the intrinsic properties of the coupled inductance, $M - L < 0$. When $i_{d1} > i_{d2}, V_{Z1} > V_{Z2}$, there are $\Delta i > 0, \frac{d\Delta i}{dt} < 0$, and the coupling inductance suppresses the mismatch current with a suppression rate of $|\frac{V_{Z1} - V_{Z2}}{M - L}|$. When $i_{d1} < i_{d2}, V_{Z1} < V_{Z2}$ with $\Delta i < 0, \frac{d\Delta i}{dt} > 0$, the coupled inductance will suppress the mismatch current, and the suppression rate is still $|\frac{V_{Z1} - V_{Z2}}{M - L}|$. When $i_{d1} > i_{d2}, V_{Z1} < V_{Z2}$ with $\Delta i > 0, \frac{d\Delta i}{dt} > 0$, the coupling inductance will increase the mismatch current to equalize the voltage drop of the two branches at a rate of $|\frac{V_{Z1} - V_{Z2}}{M - L}|$. When $i_{d1} < i_{d2}, V_{Z1} > V_{Z2}$ with $\Delta i < 0, \frac{d\Delta i}{dt} < 0$, the coupling inductance increases the mismatch current to equalize the voltage drops of the two branches, and the rate of increase remains $|\frac{V_{Z1} - V_{Z2}}{M - L}|$.

The difference between a flux cancellation coupled inductance and a flux mutual coupled inductance is in the polarity of the mutual inductance. The two-branch voltage relationships of flux cancellation coupled inductance are shown in (25) and (26), the voltage drop and self-inductance voltage relationships are shown in (27) and (28), and the mutual inductance voltage drop is different from that of flux mutual coupling, which can be expressed as (35).

$$\begin{cases} V_{M12} = M_{12} \dfrac{di_{d2}}{dt} \\ V_{M21} = M_{21} \dfrac{di_{d1}}{dt} \end{cases} \tag{35}$$

The solution process is the same as the flux mutual approach, which will not be repeated in this paper, and the obtained mismatch current transformation rate can be expressed as (36)

$$\frac{d\Delta i}{dt} = \frac{d(i_{d1} - i_{d2})}{dt} = \frac{V_{Z1} - V_{Z2}}{-(M + L)} \tag{36}$$

In the flux cancellation, when $i_{d1} > i_{d2}, V_{Z1} > V_{Z2}$, there are $\Delta i > 0, \frac{d\Delta i}{dt} < 0$, and the coupled inductance suppresses the mismatch currents with a suppression rate of $|\frac{V_{Z1} - V_{Z2}}{M + L}|$. When $i_{d1} < i_{d2}, V_{Z1} < V_{Z2}$ with $\Delta i < 0, \frac{d\Delta i}{dt} > 0$, the coupled inductance will suppress the mismatch current, and the suppression rate is still $|\frac{V_{Z1} - V_{Z2}}{M + L}|$. When $i_{d1} > i_{d2}, V_{Z1} < V_{Z2}$ with $\Delta i > 0, \frac{d\Delta i}{dt} > 0$, the coupling inductance will increase the mismatch current to equalize the voltage drop of the two branches at a rate of $|\frac{V_{Z1} - V_{Z2}}{M + L}|$. When $i_{d1} < i_{d2}, V_{Z1} > V_{Z2}$, there are $\Delta i < 0, \frac{d\Delta i}{dt} < 0$, the coupling inductance will increase the mismatch current to realize the voltage drop of the two branches are equal, and the rate of increase is still $|\frac{V_{Z1} - V_{Z2}}{M + L}|$.

To summarize the above, the coupled inductance scheme mismatch current to the two-branch MOSFET on the voltage drop is equally as critical when the two-way MOSFET current size and the voltage drop size trend are the same, which suppresses the mismatch current; when the two-way MOSFET current size and the voltage drop size of the opposite, the mismatch current is increased. The unbalanced voltage drop is suppressed, centered around the two MOSFET voltage drops being equal, and the suppression speed is $|\frac{V_{Z1} - V_{Z2}}{M - L}|$ in the flux mutual and $|\frac{V_{Z1} - V_{Z2}}{M + L}|$ in the flux cancellation.

### 3. Coupled Inductance Magnetic Device Design

The parameters of the converter used are shown in Table 1. Since the conventional transformer design is familiar to the electrical engineer, this paper only explains the coupled inductance design process.

**Table 1.** The parameters of the converter.

| Characters | Value |
|---|---|
| Input voltage $V_{in}$ | 100 V |
| Output voltage $V_{out}$ | 12 V |
| Output current $I_o$ | 10A |
| Single branch current $i_{single}$ | 5 A |
| Output voltage ripple $V_{ripple}$ | 0.5 V |
| Output capacitance $C_o$ | 470 μF |
| Single branch filter inductance $L_{single}$ | 2.2 μH |

*3.1. Magnetically Integrated Flux Analysis*

The core's magnetic flux is calculated in flux mutual and flux cancellation, and the effect of the two cases on the magnetic flux is analyzed. According to the introduction of magnetic device requirements, the parameters related to the converter core selected in this chapter are shown in Figure 6. The lengths are shown in millimeters (mm).

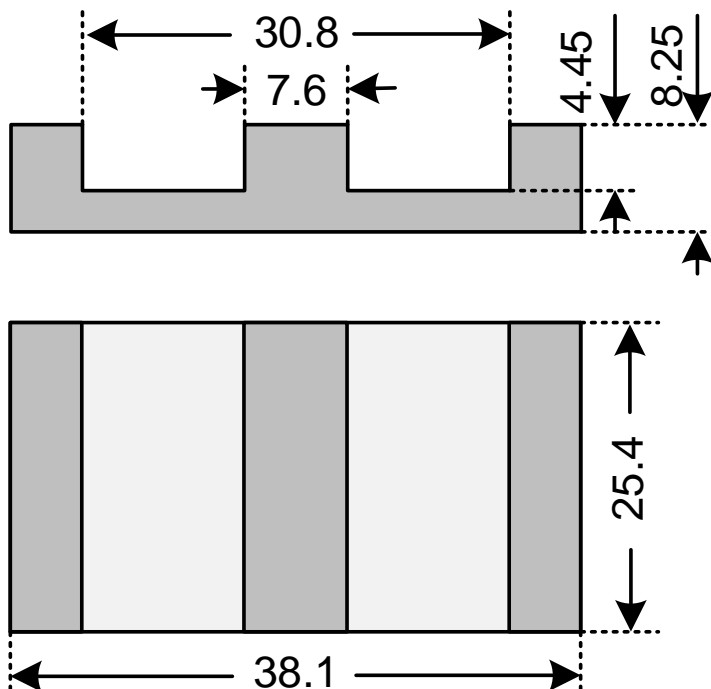

**Figure 6.** The core of the coupled inductance.

Based on the core, the coupled inductance is designed. The current direction and equivalent flux in the core are shown in Figure 7. The flux mutual forms the same direction in the core, superimposed upon each other; the flux cancellation forms the opposite direction of flux, and cancel each other. There is almost no energy stored in the flux cancellation.

According to the above flux analysis, it can be seen that the flux mutual needs the core size to meet the energy storage, while the flux mutual and flux cancellation offset one another, which results in the core size being smaller. The flux cancellation impact on the flux distribution of the core is very small, and there is an opportunity to realize integration with the main transformer.

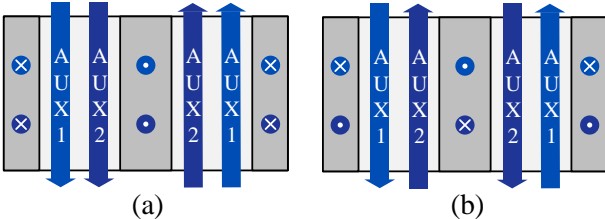

**Figure 7.** The analysis of equivalent flux: (**a**) flux mutual; and (**b**) flux cancellation.

The flyback converter working process is the first half cycle core energy storage, and the second half cycle of the core stored energy is released to the output. The coupled inductance is only related to the second half cycle. The flux distribution state of the magnetic core during the second half cycle is analyzed. The output current and flux distributions of the two branches are shown in Figure 8. The flux-coupled filter inductance of the converter is integrated into the transformer core to realize the double utilization of the core.

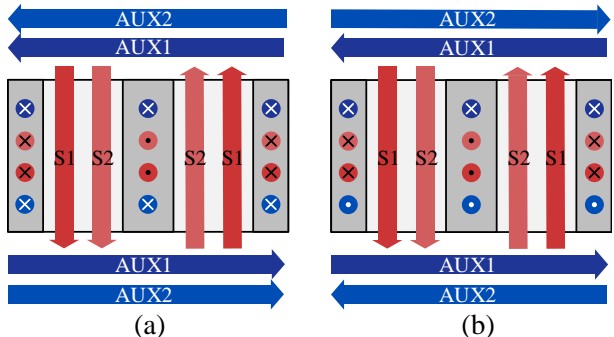

**Figure 8.** The magnetic integration analysis: (**a**) flux mutual; and (**b**) flux cancellation.

As shown in Figure 8, the two secondary currents are donated by S1 and S2, so the two filter currents are denoted by AUX1 and AUX2, respectively. The coils of the coupled inductance surround the core, and the entire core and the peripheral air form a closed flux loop with a low coupling coefficient. The two auxiliary coils generate the magnetic flux on the core in flux mutual in the same direction while generating flux cancellation in the opposite direction.

Since the two filter currents are equal in magnitude and arranged in the same way in mutual cancellation, the flux generated by the two auxiliary coils can cancel each other out. Based on this, the coupled coils do not affect the flux distribution, and the auxiliary coils can be integrated into the main core. The integrated strategy reduces the filter core and improves the power density of the converter.

### 3.2. The Simulation of the Magnetic Integration

To verify the analysis results, the integrated magnetic device is modeled in the MAXWELL module of ANSYS software, and the coupling coefficients of the primary winding, secondary winding and auxiliary winding are simulated and analyzed. The model in MAXWELL is shown in Figure 9.

The winding and core relationship is schematically shown in the front view section in Figure 10. To improve the coupling coefficient between the primary and secondary sides as well as the consistency between the two windings of the secondary side, the secondary1 and secondary2 windings are arranged in symmetrical positions above and below the primary winding. The two auxiliary windings are on top of the secondary winding. The primary winding, the secondary1 and secondary2 windings as well as the auxiliary1 and auxiliary2 windings and the isolation medium FR4 are represented. The auxiliary and main windings are discrete monolithic structures that realize the disassembly and assembly of the magnetic integration.

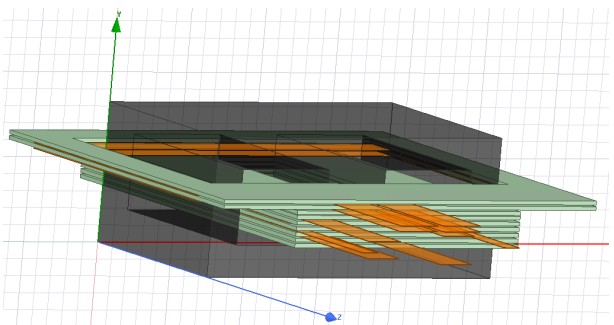

**Figure 9.** The integrated magnetic device model.

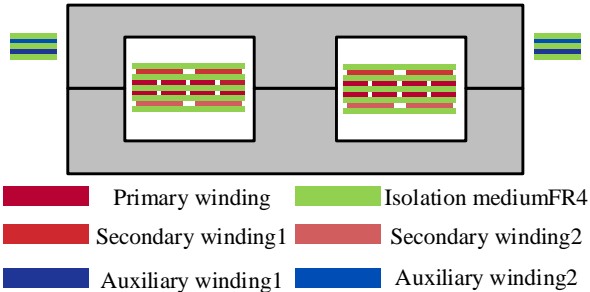

| | |
|---|---|
| ▮ Primary winding | ▮ Isolation mediumFR4 |
| ▮ Secondary winding1 | ▮ Secondary winding2 |
| ▮ Auxiliary winding1 | ▮ Auxiliary winding2 |

**Figure 10.** The  front cutaway view of the integrated magnetic device model.

The winding width, thickness, and other related parameters are shown in Table 2. Since the secondary and auxiliary winding currents are equal, the winding width is set as equal. Based on the manufacturing cost and on-resistance relationship, a 2 ounce copper thickness was selected, and its thickness is 70 μm.

**Table 2.** The parameters of the transformer's windings.

| Winding | Width | Copper Thickness |
|---|---|---|
| Primary winding | 2 mm | 70 μm |
| Secondary winding1 | 4.2 mm | 70 μm |
| Secondary winding2 | 4.2 mm | 70 μm |
| Auxiliary winding1 | 4.2 mm | 70 μm |
| Auxiliary winding1 | 4.2 mm | 70 μm |

Based on the arrangement of the windings, the transformers without/with auxiliary windings were simulated separately to form a comparison. The simulation flux distributions of the main transformer without/with auxiliary windings are shown in Figure 11. Comparing the flux distributions between the transformer with and without auxiliary windings, the magnetic flux is essentially the same. The coupling coefficients are derived from flux-related data for quantitative analysis. The coupling coefficients of the transformers without/with auxiliary windings are listed in Tables 3 and 4, respectively.

Comparing the coupling coefficients without/with auxiliary windings, the coupling coefficients between the primary and secondary windings increase from 0.991 to 0.992. Since the changes in the coupling coefficients are small enough, the effect can be neglected. The coupling coefficient between the two auxiliary windings is 0.150. From (36), the 0.150 coupling coefficient can work effectively in the flux cancellation application.

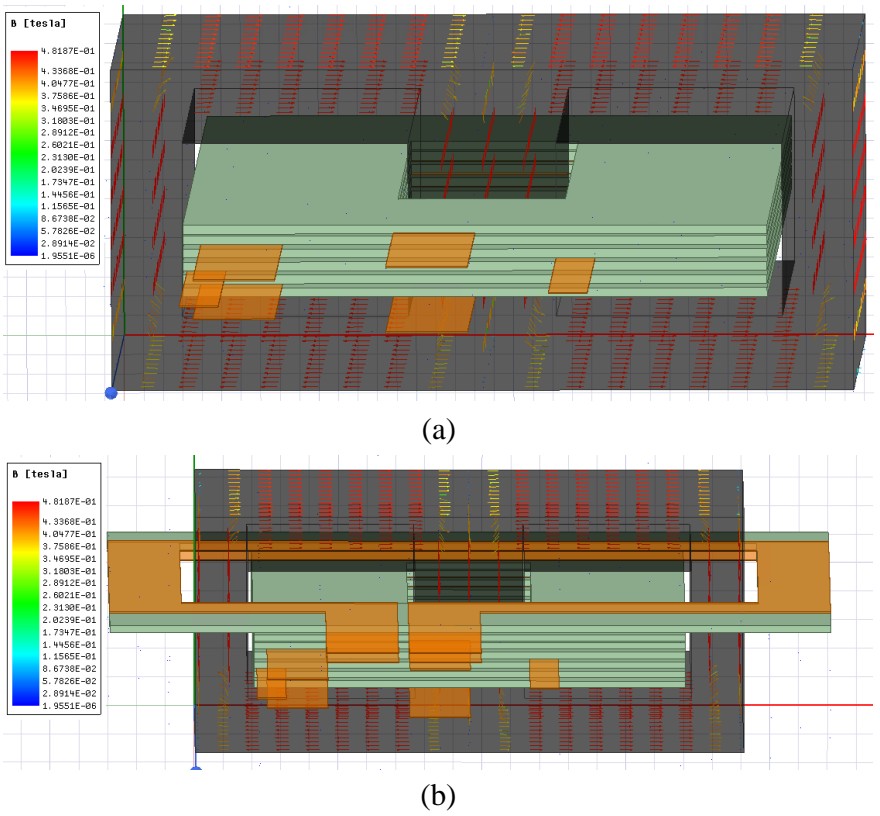

(a)

(b)

**Figure 11.** The simulation flux distributions of the main transformer without/with magnetic integration: (**a**) without magnetic integration; and (**b**) with magnetic integration.

**Table 3.** The simulation results of the coefficients of the transformer without the auxiliary windings.

|  | **Primary** | **Secondary1** | **Secondary2** |
|---|---|---|---|
| Primary | 1 | 0.991 | 0.991 |
| Secondary1 | 0.991 | 1 | 0.984 |
| Secondary2 | 0.991 | 0.984 | 1 |

**Table 4.** The simulation results of the coefficients of the transformer with the auxiliary windings.

|  | **Primary** | **Secondary1** | **Secondary2** | **Auxiliary1** | **Auxiliary2** |
|---|---|---|---|---|---|
| Primary | 1 | 0.992 | 0.992 | 0.030 | 0.027 |
| Secondary1 | 0.992 | 1 | 0.984 | 0.048 | 0.027 |
| Secondary2 | 0.992 | 0.984 | 1 | 0.031 | 0.024 |
| Auxiliary1 | 0.030 | 0.048 | 0.031 | 1 | 0.150 |
| Auxiliary2 | 0.027 | 0.027 | 0.024 | 0.150 | 1 |

## 4. Experimental Results

### 4.1. Verification of the Magnetic Integrated Transformer

The magnetically integrated layout was designed, fabricated and tested according to the magnetically integrated design scheme, in which the auxiliary and main windings are discrete monolithic types. The physical photos of the main and auxiliary windings and the integrated transformer are shown in Figure 12. The primary winding ports are on its left side, whilst the two secondary side windings are distributed in the two ports above and below the winding, respectively. The auxiliary windings have a total of six ports, comprising the upper three ports which correspond to the secondary1 winding and the lower three ports which correspond to the secondary2 winding.

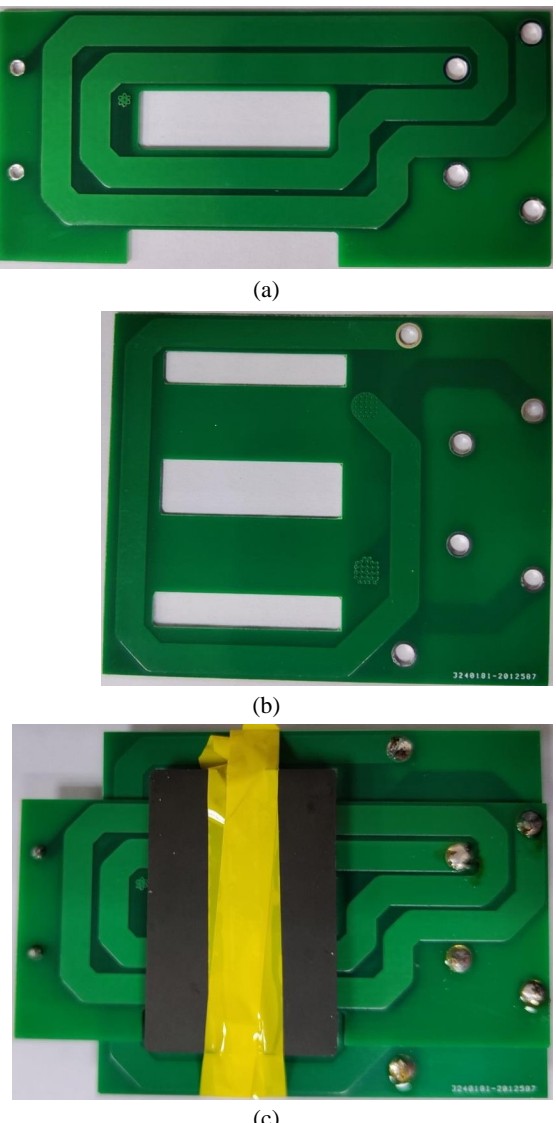

**Figure 12.** The experimental prototype of the integrated transformer: (**a**) the main windings; (**b**) the auxiliary windings; and (**c**) the integrated transformer.

The coupling coefficients of the integrated transformer are measured to verify the simulation results, and the test results are shown in Table 5.

Comparing the simulation and test coupling coefficients, the coupling coefficients of the primary and secondary windings in the test results are higher than the simulation results by about 0.006. According to the definition of the coupling coefficients, their influence on the converter's performance is very small and can be ignored. The test coupling coefficient between the two auxiliary windings is higher than the simulation result. This is because ANSYS uses the finite element simulation method, its simulation area is the air box that is immediately adjacent to the auxiliary winding, and the magnetic circuit formed by the air around the auxiliary winding is calculated. The simulation results of the transformer and the test results match well with the key parameters.

**Table 5.** The tested results of the coefficients of the transformer with the auxiliary windings.

|  | **Primary** | **Secondary1** | **Secondary2** | **Auxiliary1** | **Auxiliary2** |
|---|---|---|---|---|---|
| Primary | 1 | 0.998 | 0.998 | 0.027 | 0.025 |
| Secondary1 | 0.998 | 1 | 0.994 | 0.021 | 0.026 |
| Secondary2 | 0.998 | 0.994 | 1 | 0.025 | 0.022 |
| Auxiliary1 | 0.027 | 0.021 | 0.025 | 1 | 0.210 |
| Auxiliary2 | 0.025 | 0.026 | 0.022 | 0.210 | 1 |

*4.2. Verification of Suppression Strategy*

The proposed suppression is verified using SiC diodes as rectifier devices. The primary switch of the converter adopts the CPM309000065B SiC MOSFET from CREE. In general, the diode mismatch is relatively small, the mismatch characteristics are difficult to test and demonstrate. The more severe test conditions were constructed to verify the effectiveness of this strategy. The rectifier diodes on the secondary side are selected as the ASD10120C SiC diode from AnBon, and MUR1520 Si diode from Onsemi. The specific parameters are shown in Table 6.

**Table 6.** The parameters of the diodes used.

| **Characteristics** | **ASD10120C** | **MUR1520** |
|---|---|---|
| Repetitive Peak Reverse Voltage | 1200 V | 200 V |
| Forward Voltage @ 10 A | 1.6 V | 0.9 V |
| Average Rectified Forward Current | 29 A | 15 A |

A comparison group without auxiliary windings was constructed to verify the effectiveness of the magnetic integration transformer, and the schematic of the experimental prototypes and the prototypes themselves are shown in Figures 13 and 14, respectively.

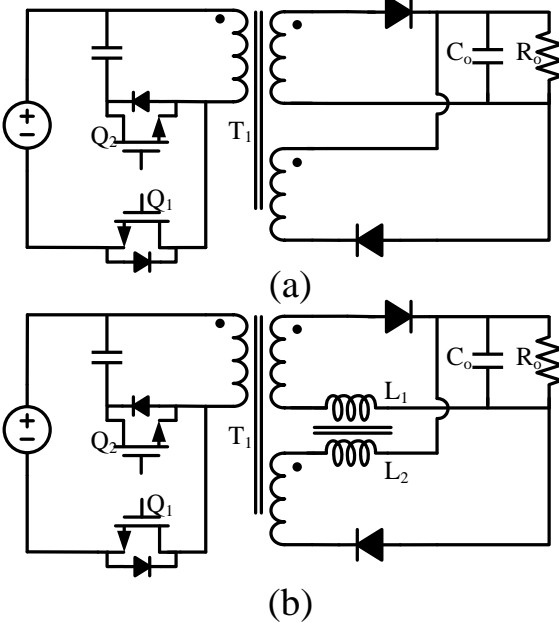

(a)

(b)

**Figure 13.** The schematic of the experimental prototypes: (**a**) without magnetic integration; and (**b**) with magnetic integration.

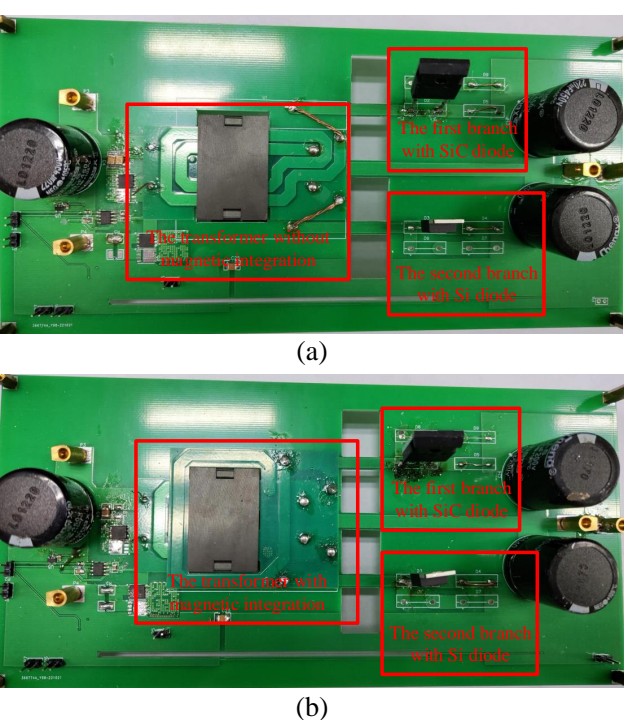

**Figure 14.** The experimental prototypes: (**a**) without magnetic integration; and (**b**) with magnetic integration.

The experimental principle prototypes use ASD10120C in the first branch and MUR1520 in the second branch to achieve the mismatch of the two branches. The test results in different loads without/with magnetic integration are shown in Figure 15.

The three test lines indicate the diode voltage drop of the first branch, the diode voltage drop of the second branch, and the difference in the voltage drop between the two branches, respectively. Comparing the test results without/with magnetic integration in the same load, the voltage difference between the two branches of the experimental prototype with the magnetic integration is significantly smaller. Among them, the second branch test waveform of the prototype without magnetic integration shows drastic voltage fluctuations during the diode conduction process. The test results reflect that the two branches have a large mismatch without magnetic integration and the second branch is unstable. There is a mismatch in the voltages of the two branches of about 0.1 V on the experimental prototype with the magnetic integration under all load conditions, which is the result of the mismatch between the coupled inductance of the two branches. Due to the actual manufacturing process, the main winding PCB has a notch in the lower part of the main winding, which has a certain effect on the self-inductance of the secondary main winding, and in the mismatch suppression process, there is a small voltage difference of 0.1 V on the diode to match the mismatch of the secondary winding of the two branches. The consistency of the voltage drop difference under different loads verifies the analysis. The stable operation under different load cases verifies the stability of the magnetic integration control method proposed in this paper. The converter has a stable mismatch rejection capability under different load cases. This is consistent with the previous analysis that the coupled inductance suppresses the voltage drop mismatch reduction direction of the two branches.

According to the operation of the converter, the effect of the mismatch of the two branches on the core bias is tested. After 5 min of steady-state operation with a 4 A load, the thermal distribution of the core and its windings was stable and photographed. The thermal distribution is shown in Figure 16.

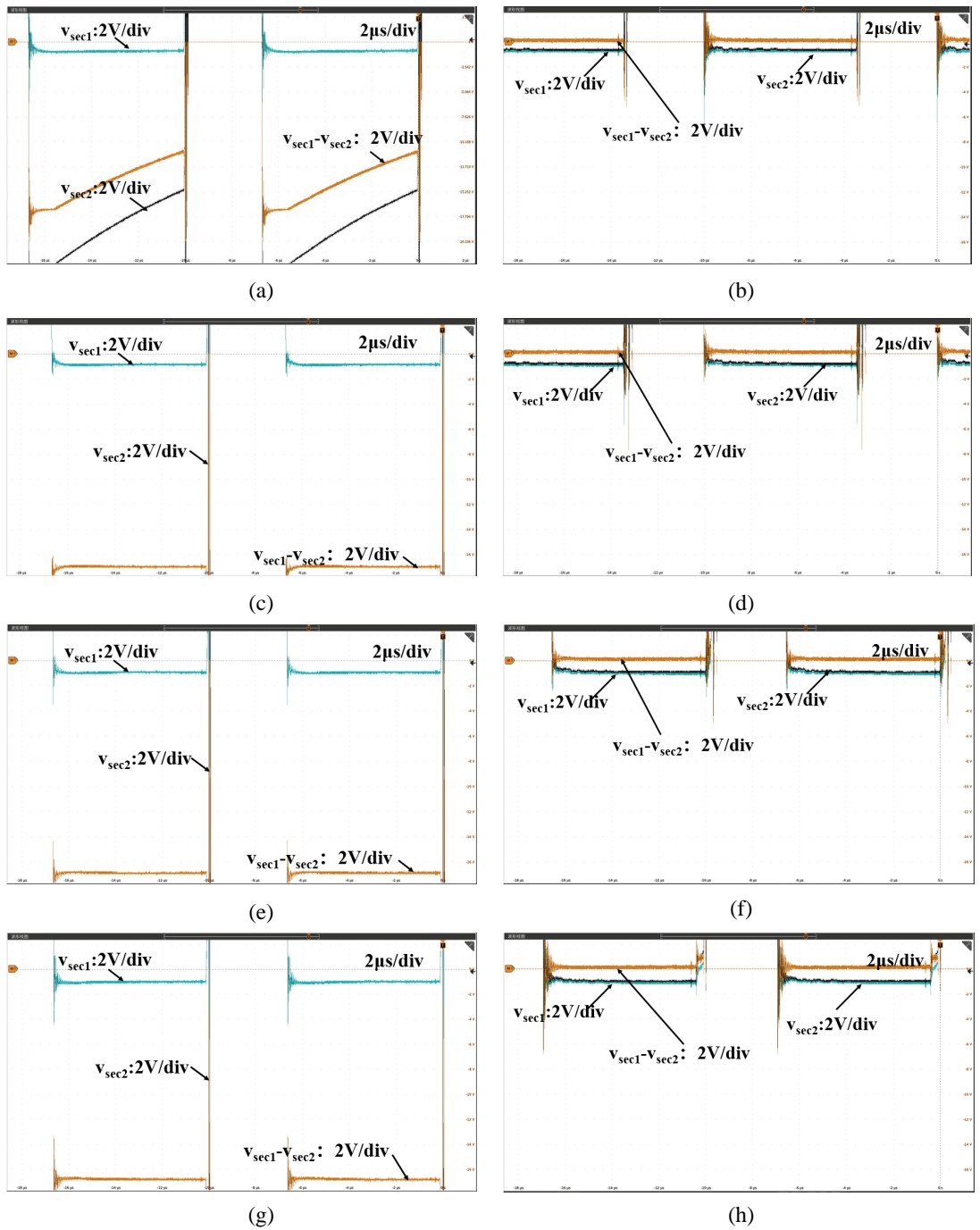

**Figure 15.** The voltage drop characteristics of the two branches: (**a**) 1 A load without magnetic integration; (**b**) 1 A load with magnetic integration; (**c**) 4 A load without magnetic integration; (**d**) 4 A load with magnetic integration; (**e**) 7 A load without magnetic integration; (**f**) 7 A load with magnetic integration; (**g**) 10 A load without magnetic integration; (**h**) 10 A load with magnetic integration.

In the case of the 4 A load, the maximum temperature of the transformer without core integration is 39.3 °C, the average temperature is 37.6 °C, the maximum temperature of the transformer with magnetic integration is 35.8 °C, and the average temperature is 34.1 °C. The overall operating temperature of the transformer with magnetic integration is significantly lower than that of the transformer without magnetic integration. The temperature distribution diagram shows that the magnetic integration scheme has little effect on the magnetic energy distribution of the core, and its suppression of current loss

can effectively reduce the working temperature of the transformer, which has a good effect on the thermal distribution and reliability of the converter.

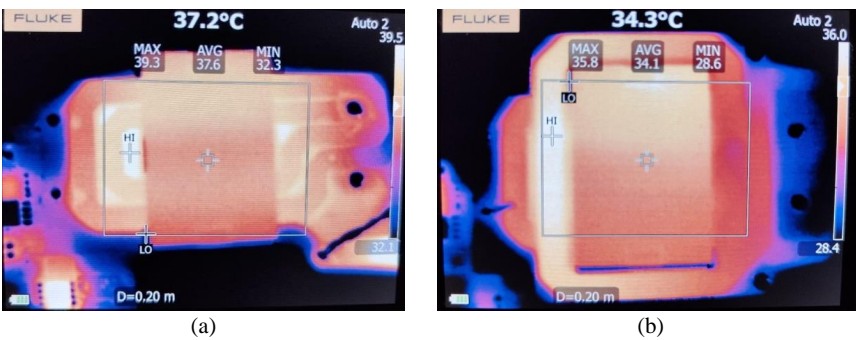

(a)  (b)

**Figure 16.** The temperature distribution of the transformer: (**a**) without magnetic integration; and (**b**) with magnetic integration.

According to the test waveform and the calculation results of the mismatch current, it can be seen that the prototype of the magnetic integration principle can effectively suppress the current mismatch caused by the mismatch. The efficiency test with/without magnetic integration is carried out under different load conditions, and the efficiency test results are shown in Figure 17.

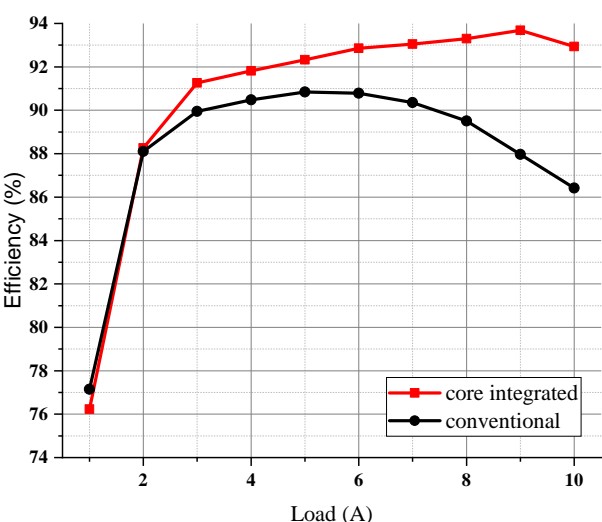

**Figure 17.** The efficiency comparison.

When the load is 1 A, the efficiency of the converter with magnetic integration is lower than that of the traditional converter, which is due to the new auxiliary winding increasing the copper loss of the transformer. This part of the conduction loss accounts for a large proportion in light load, which reduces the efficiency of the principle prototype with magnetic integration. With the load increasing, the efficiency of the converter with magnetic integration monotonically increases in the load range of 1–9 A, while the efficiency of the traditional converter only increases in the range of 1–5 A. With the load increasing, the current is distributed in the two branches, and the diode conduction and reverse recovery loss is small, while the diode conduction loss of the traditional converter rapidly increases due to the uneven current distribution. The efficiency of the converter without magnetic integration decreases after the 5 A load because the diode conduction and reverse recovery loss account for the loss dominance. As the load continues to increase, the efficiency of the magnetic integration prototype reaches its maximum at 9 A, which is 93.68%. Through the above efficiency analysis, it can be seen that the magnetic integrated control method

can effectively improve the efficiency of the current offset converter, especially under the condition of heavy load, as the efficiency of the converter is more obvious.

The efficiency improvement of this strategy was tested at different temperatures, and the test results are shown in Figure 18. This strategy worked effectively in the range from −25 °C to 75 °C. The enhancement effect is more obvious under heavy-load and high-temperature conditions. This is because the on-state voltage drops of Si and SiC diodes become opposites as the operating temperature increases. The on-state voltage of the Si diode drops while the SiC diode rises as the operating temperature increases, which leads to a more serious mismatch in the conventional method. The mismatch reduces the efficiency of the conventional converter. The proposed strategy suppresses the mismatch, and the efficiency improvement is more obvious.

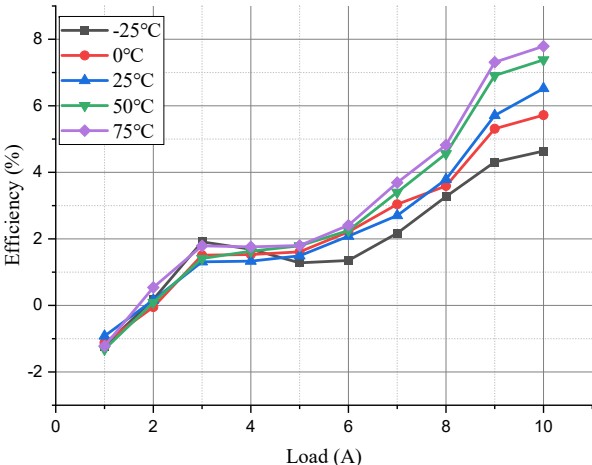

**Figure 18.** The efficiency improvements at different temperatures.

For long-term stability and effectiveness, the life test was performed. The principle prototype operated continuously for 10 h with 10 A load at room temperature. The efficiency variations are shown in Figure 19. The efficiency degradation is within 3%, which is acceptable in commercial applications.

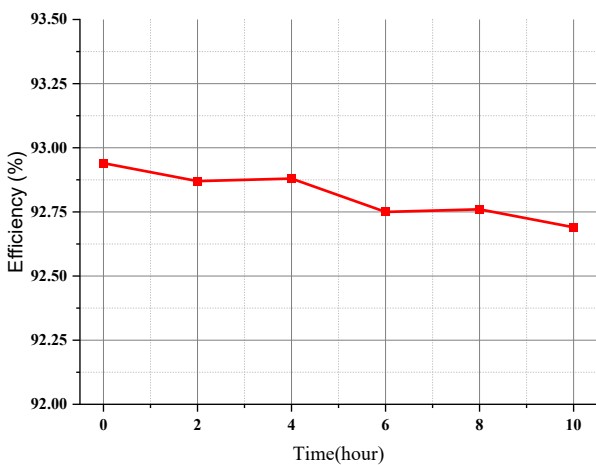

**Figure 19.** The efficiency variations in long-term operating.

## 5. Conclusions

In high-frequency applications, the mismatch between devices and circuits can have extremely harsh effects. To take full advantage of the high-frequency characteristics of SiC power devices, suppressing the mismatch has become an urgent problem. Starting from the current mismatch problem of parallel devices, this paper analyzes the working

mechanism of the coupling inductance to suppress the mismatch and reveals that the mismatch suppression of the coupling inductance on different branches is essentially the essence of the suppression of the voltage mismatch of different branches. Furthermore, a novel magnetic integration strategy is proposed to suppress the mismatch. The strategy integrates the coupling inductance into the main core and balances the mismatch of the two branches at the output end by suppressing the voltage difference. To verify the control method, the design and manufacture of the experimental prototype were carried out, and the effectiveness of the control method was verified in the whole load by constructing severe mismatch conditions. The mismatch voltage of the two branches is controlled within 0.1 V. Compared with the comparison group, the proposed strategy suppresses the loss caused by mismatch and improves the efficiency of the converter. The efficiency of the magnetic integrated converter at full load is 92.94%, which is 6.52% higher than that of the traditional converter, and the prototype of the magnetic integration principle has a maximum efficiency of 93.68% at a load of 9 A. The technical advantages are analyzed above, whilst the scalability and mass production are analyzed as follows. The principle prototype produced in this paper is based on a commercial printed-circuit-board (PCB) preparation process and mature commercial devices, so it is a perfect match for existing technological fabrication processes. The magnetic integration strategy reduces an auxiliary winding core, and uses separated windings to avoid increasing the number of layers of PCB, which effectively controls the cost of the converter. Therefore, mass production is not a problem in terms of manufacturing and cost.

**Author Contributions:** Conceptualization, L.C. and Z.L.; methodology, S.S.; software, J.L.; validation, J.L., W.Y. and Y.S.; formal analysis, H.G.; investigation, H.G.; resources, Y.W.; data curation, S.S.; writing—original draft preparation, H.G.; writing—review and editing, Y.W.; visualization, Y.W.; supervision, L.C.; project administration, Z.L.; funding acquisition, L.C. All authors have read and agreed to the published version of the manuscript.

**Funding:** The APC was funded by National High-Level Personnel of Special Support Program (2023) and Independent Research and Development Program of CASC (YJ2345CX).

**Data Availability Statement:** The authors confirm that the data supporting the findings of this study are available within the article.

**Conflicts of Interest:** The authors declare no conflicts of interest.

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
