# Peer review of "A Magnetic Integration Mismatch Suppression Strategy for Parallel SiC Power Devices Applications"

_electronics, doi:10.3390/electronics13050954_

Round 1
Reviewer 1 Report
Comments and Suggestions for Authors
This manuscript presents an innovative strategy for suppressing magnetic integration mismatch in parallel Silicon Carbide (SiC) power devices, focusing on the utilization of coupled inductance to address current imbalance. The method, supported by theoretical analysis and experimental results, shows potential in enhancing efficiency and voltage mismatch suppression in power electronics. I am favorable to the publication of this manuscript and recommend the following enhancements to further strengthen the paper:
1. Enhanced Experimental Validation: Conduct additional experiments under varied operational conditions, such as different temperatures and load variations, to demonstrate the robustness of the proposed strategy. A comparative analysis against standard configurations under these conditions would underscore its practical applicability.
2. Long-term Stability and Effectiveness: Include a detailed discussion on the long-term stability of the method, considering thermal stability and aging effects. Use accelerated life testing to simulate extended usage and observe any efficiency loss or degradation.
3. Scalability and Commercial Application: Discuss the scalability challenges for mass production, including manufacturing complexities and cost implications. Provide a feasibility analysis or case study illustrating the integration of this strategy into existing manufacturing processes and its economic and technical advantages in a commercial setting.
Reviewer 2 Report
Comments and Suggestions for Authors
This article introduces an innovative approach to integrating mismatched SiC power devices, ensuring uniform operating conditions. The paper is comprehensive, encompassing three key aspects:
1. A theoretical analysis of magnetic coupling in a circuit that includes a MOS transistor.
2. Simulation of the designed transformer.
3. Experimental validation of the proposed solution through precise measurements.
However, there are several areas where the article could be improved for clarity and completeness:
- On page 9, Figure 6 displays numerals in a font size that significantly differs from other graphs, impacting readability.
- In Table 2 on page 11, the data for "Copper thickness" in the last column is notably absent.
- The scale numbers in Figure 11 on page 12 are nearly indecipherable.
- On page 15, the axis numbers in Figure 14 are difficult to read.
- There is a potential error in Figure 16 on page 17; the caption reads "The heat distribution…", which requires verification.
- On line 368, the reference "Fig. refEfficiencyheat" appears to be incorrectly formatted.
- Line 145 contains an anomalous "m_blacksquare" character.
Additionally, I have several questions to further understand the study:
1. Could the schematic for the experimental prototypes illustrated in Figure 13 be provided?
2. The article discusses measurements on SiC diodes, yet the theoretical analysis focuses on MOSFETs. What is the rationale behind this discrepancy?
3. What was the temperature of the diodes under maximum load (10A)?
4. Were any heat sinks employed for the power components? If not, do you believe their inclusion would alter the efficiency of the system?
